# Multi-Functional Properties of MWCNT/PVA Buckypapers Fabricated by Vacuum Filtration Combined with Hot Press: Thermal, Electrical and Electromagnetic Shielding

**DOI:** 10.3390/nano10122503

**Published:** 2020-12-14

**Authors:** Liyang Cao, Yongsheng Liu, Jing Wang, Yu Pan, Yunhai Zhang, Ning Wang, Jie Chen

**Affiliations:** 1Science and Technology on Thermostructural Composites Materials Laboratory, Northwestern Polytechnical University, Xi’an 710072, China; liyangcao@mail.nwpu.edu.cn (L.C.); wangjing1@nwpu.edu.cn (J.W.); zhyhai@mail.nwpu.edu.cn (Y.Z.); mugglejie@mail.nwpu.edu.cn (J.C.); 2NPU-SAS Joint Research Center of Advanced Ceramics, Northwestern Polytechnical University, Xi’an 710072, China; 3Science and Technology on Scramjet Laboratory, College of Aerospace Science and Engineering, National University of Defense Technology, Changsha 410073, China; clyamu@foxmail.com

**Keywords:** MWCNT buckypaper, vacuum filtration, hot press, electrical conductivity, thermal conductivity, finite element

## Abstract

The applications of pure multi-walled carbon nanotubes (MWCNTs) buckypapers are still limited due to their unavoidable micro/nano-sized pores structures. In this work, polyvinyl alcohol (PVA) was added to a uniform MWCNTs suspension to form MWCNT/PVA buckypapers by vacuum infiltration combined with a hot press method. The results showed an improvement in the thermal, electrical, and electromagnetic interference (EMI) shielding properties due to the formation of dense MWCNTs networks. The thermal and electrical properties rose from 1.394 W/m·k to 2.473 W/m·k and 463.5 S/m to 714.3 S/m, respectively. The EMI performance reached 27.08 dB. On the other hand, ABAQUS finite element software was used to simulate the coupled temperature-displacement performance. The electronic component module with buckypapers revealed a homogeneous temperature and thermal stress distribution. In sum, the proposed method looks promising for the easy preparation of multi-functional nanocomposites at low-cost.

## 1. Introduction

Electronic products, such as smartphones, tablet computers and laptops are currently developing at an accelerated pace. Such features require hardware that has a great electrical performance, heat transfer efficiency, and that is lightweight [1]. Meanwhile, electromagnetic (EM) waves pollution generated from these devices could be harmful to human health and should be isolated and limited [2,3,4,5]. Hence, it is imperative to develop multifunctional materials that meet such application requirements.

Carbon nanotubes have attracted increasing attention since the 1990s due to their excellent mechanical, electrical, and thermal properties [6,7,8]. Note, that carbon nanotubes paper or buckypaper is made from carbon nanotubes [9,10]. On the other hand, single carbon nanotubes have excellent electrical and thermal conductivities, which theoretically could reach 190 k/μm and 3000 W/m k, respectively. Such functional properties would remarkably be reduced due to crookedness and agglomeration of CNTs, leading to the formation of numerous pores and interfaces [11,12,13].

Several methods have so far been developed to fabricate well-organized single-wall carbon nanotubes (SWCNTs) or multi-walled carbon nanotubes (MWCNTs) buckypapers. The growth of aligned CNTs on templates by the chemical vapor deposition (CVD) method has led to the formation of super-aligned buckypaper with improved thermal conductivity up to 766 ± 77 W/m·k [1]. Lee et al. [14] applied a dip-coating process to prepare silver nanowire/carbon nanotube/cellulose hybrid papers with an electromagnetic interference (EMI) shielding effectiveness of 23.8 dB and enhanced electrical conductivity from 0.22 S/cm to 2.83 S/cm. Shashikant et al. produced polyvinyl alcohol (PVA) incorporated CNT-graphene composites with an electrical conductivity of 82 S/cm. Bradford et al. [15] utilized the shear pressing method to process tall vertically aligned CNT arrays grown on a Si wafer into dense aligned CNT preforms and then composites. Trakakis et al. [16] fabricated strong multi-walled carbon nanotube/epoxy composites and found that process parameters, such as grafting functionalities onto the CNT surface, CNT alignment (for longer nanotubes), buckypaper porosity, and prepregging at high pressures, were significant for the production of lightweight composites. In another piece of work, Trakakis et al. [17] modified CNTs through epoxidation treatment to achieve high-degree chemical bonding between the epoxy matrix and a high volume fraction of CNTs. However, the preparation processes mentioned above are still limited in terms of time-consumption, high cost of experimental facilities, and uncontrollability over the reactions. This, in turn, would limit the practical applications of buckypapers [18,19,20,21]. In addition, these techniques have not been focused on reducing porosity within buckypapers. Compared to above methods used for the fabrication of buckypapers, vacuum filtration is widely used thanks to its convenience and efficiency. Nevertheless, pure buckypapers are less useful for engineering applications due to their high friability and poor functional properties. To overcome the shortcomings of buckypapers, some strategies have been adopted to prepare buckypapers composites, including buckypaper/polymer [22,23], buckypaper/graphene [24,25,26], and buckypaper/metallic oxide [27,28,29,30]. On the other hand, hot press is another common, low cost and high-efficiency method, often employed in the curing process of polymer and ceramic materials [29,31,32,33,34]. However, so far, very few studies have analyzed the relationships between multifunctional properties, CNT networks, and preparation methods.

PVA is a water-soluble, non-poisonous and biocompatible polymer material widely used in composites. PVA is characterized by film-forming properties, largely expending its application fields to medical equipment, blend membrane, and textile industry. Recently, some studies have been focused on the preparation of CNT/PVA composites with random or aligned CNT arrays. Most of these studies have dealt with evaluation of the mechanical properties, such as tensile strength (19.3 ± 1.11 MPa), fracture toughness (712 ± 69.8 KJ m^−3^), and Young’ modulus (1273 ± 56.1 MPa) [35,36,37].

In this work, a vacuum filtration-hot press method was employed to fabricate MWCNT/PVA buckypapers with controlled PVA content. The electrical, thermal, and EMI shielding performances were measured and reinforcement mechanisms behind such functional properties of MWCNT/PVA buckypapers were also investigated. The relationships between the microstructure and multifunctional properties of buckypapers were revealed. In addition, the finite element analysis software ABAQUS was employed to highlight the coupled temperature-displacement characteristics of electronic components with buckypapers.

## 2. Materials and Methods

### 2.1. Materials

Multi-walled carbon nanotubes (MWCNTs) with outer diameters from 8 to 15 nm and length from 10 to 50 μm were supplied by Chengdu Organic Chemicals Co, Ltd (Chengdu, China). Poly (vinyl alcohol) 1799 (PVA1799, 90%) was purchased from Shanghai Macklin Biochemical Co, Ltd (Shanghai, China). Nitric acid (65–68%) was obtained from Chengdu Chron Chemical Co, Ltd (Chengdu, China).

### 2.2. Preparation of MWCNT/PVA Buckypaper

First, the MWCNTs were dispersed in 250 mL of HNO_3_ at 80 °C for 1 h. Then, the mixture was dialyzed using a dialysis membrane and immersed in deionized water, where the pH value was monitored. After 24 h, the neutral solution (pH = 7) was filtered off and functionalized MWCNTs were washed and vacuum dried for at 80 °C for 24 h. Second, 0.5 g MWCNTs were added into 250 mL deionized water followed by sonication in a bath for 20 min to yield well-dispersed MWCNTs suspension. Amounts of 0.25 g, 0.1 g, 0.2 g and 0.3 g PVA were then added into MWCNT suspensions followed by heating at 95 °C for 60 min. Next, vacuum filtration-hot press method was used to form MWCNT/PVA buckypapers. The obtained MWCNTs solution was then filtrated through filter membrane (PTFE, 0.45 μm, φ10 cm). After solution filtration, MWCNT/PVA buckypapers were removed from the membrane, dried at 75 °C for 20 min, and hot pressed under 50 MPa at 70 °C (Figure 1a). For comparison, MWCNT/PVA buckypapers with different PVA contents without hot pressing were also fabricated following the same method. Figure 1b depicts the microstructure evolution of MWCNTs and buckypapers during the preparation process, and Figure 1c–e are photographs of the MWCNT/PVA buckypapers.

### 2.3. Characterization

Raman spectroscopy (confocal Raman microscope, Renishaw, Beijing, China) was carried out to verify the changes in graphite structure after nitration. The electrical conductivities of samples with diameter of 90 mm were obtained by the 4-point probe (RTS-8, Xi’an, China) method. Thermal conductivity was measured by a Hot-Disk thermal constant analytical instrument (TPS2200, Uppsala, Sweden). The microstructure of buckypapers was observed by scanning electron microscopy (FEI, Helios G4 CX, Waltham, MA, USA). The EMI shielding performance measurements were conducted on a vector network analyzer (VNA, MS4644A; Anritsu, Atsugi, Japan) using the waveguide (8.2–12.4 GHz) method. Hot press treatment was carried out by means of the vulcanizing press method (XLB-D, Xi’an, China). Finite element simulations of heat transfer effect of buckypapers were performed by finite element software ABAQUS. Table 1 shows the dimension parameters of pores (generated by tangled and overlapped between CNTs) with different PVA content before and after hot pressing. The obtained MWCNT/PVA buckypapers were named as BP5.7#, BP16.7#, BP28.6#, and BP37.5#, and had a PVA content in the MWCNTs/PVA solution of 5.7 wt%, 16.7 wt%, 28.6 wt%, and 37.5 wt%, respectively. For comparison, buckypapers without PVA were also prepared following the same process (named as BP0#). The parameters were repeatedly measured and scanning electron microscopy images were obtained, which were analyzed by an image-based method using the Image J software (version 1.48, National Institutes of Health, Bethesda, Rockville, MD, USA). It should be mentioned that Image J is an image processing software which has been widely used in scientific image processing. In this work, in order to prevent the destruction of thin buckypapers by the mercury intrusion method, the Image J software was used to determine the average pore diameters and areas [38,39,40].

## 3. Results

### 3.1. Raman Spectrum of Buckypapers

The purpose of the acidification treatment of MWCNTs instead of using dispersant like Triton-X was to eliminate the interference from small molecules in dispersant during the measurement of electrical, EMI, and thermal performances. In the Raman spectrum of CNTs, the D peak is mainly derived from defects and the amorphous carbon in CNTs, while the G peak is considered to be generated by two E_2g_ Raman active vibration modes, and is frequently used to describe the graphitization degree of CNTs. The relative intensity ratio (*I_D_/I_G_*) of the D and G peaks can reflect the disorder degree and defect intensity of the CNTs. In addition, the intrinsic reactivity of the sidewall carbon atoms is sensitive to the curvature of the CNTs and the density of the defects [41]. Nitric acid with strong oxidability should functionalize MWCNTs with hydrophilic groups, such as hydroxyl groups and carboxyl groups. After acid treatment, the walls of the CNTs were oxidized and etched, and the carboxyl group would have also been induced partially at the sidewall of CNTs, as well as at the ends [42], which increased the surface defects of the CNTs. Lattice distortion occurred and the spacing of the tube wall became larger. On the other hand, CNTs could be cut into short lengths by the acids. This will lead to better dispersion of the modified MWCNTs in aqueous solution. However, nitric acid could partly destroy graphitic integrity of MWCNTs, thus degrading the properties of MWCNTs. To study the influence of acid treatment, the graphite structures of MWCNTs were investigated by Raman spectroscopy. In Figure 2, all curves (including three acid treated samples and one pristine sample) exhibited typical characteristic peaks at around 1347 cm^−1^ and 1581 cm^−1^, which were attributed to the D and G bands. The intensity ratios of D to G (*I_D_/I_G_*) for all curves were estimated as 1.26, 1.34, 1.42, and 1.45. Hence, acidified MWCNTs possessed higher degrees of disorder. Both functionalization and the occurrence of defects altered the length of the atomic valence bonds, and changed the hybrid orbital type of some carbon atoms. Milowska et al. [43] used the density functional theory to investigate the morphology and structure of single-walled carbon nanotubes modified by –CH_n_, –NH_n_, –COOH, and –OH functional groups. They observed that functionalization resulted in topo-rehybridization from sp^2^ to sp^3^ in the CNTs, which led to conversion between metallic or semiconductor properties. This provides the basis for the development of composite materials, such as buckypapers with multifunctional properties.

### 3.2. Microstructure of MWCNT/PVA Buckypapers

Figure 3 shows the microstructures of MWCNT/PVA buckypapers. At 5.7 wt% (BP5.7#) PVA content, the MWCNT/PVA buckypaper showed smooth surface with less aggregated PVA when compared to buckypapers with 37.5 wt% (BP37.5#) PVA content. However, excess PVA content led to an irregular buckypaper surface due to the strong aggregation of PVA (white stripes in Figure 3b). Additionally, continuous aggregated PVA was evidently formed (Figure 3a,b). When the buckypaper was hot pressed, the connection between intertubes became tight and the number of pores with different size were closed, as well as the compressed aggregated PVA, which can be clearly observed in Figure 3c,d. On the other hand, the glass transition temperature of the PVA was determined as 75 °C [44], meaning that molecular chain segments of the PVA would move when the temperature was close to 75 °C. In the much denser CNT networks, which were generated by high pressure and temperature close to the glass transition temperature, flowable chain segments would help the redistribution of PVA and reduce agglomeration. In addition, PVA would connect the adjacent nanotubes, which would be stronger than the van der Waals forces, and would weaken the mechanical entanglements between nanotubes.

### 3.3. Thermal Conductivity and Finite Element Analysis

#### 3.3.1. Thermal Conductivity

The heat conduction mechanism of CNTs should be dominated by phonon heat conduction. Meanwhile, the contact resistance (interface resistance) plays an important role in the heat conduction of solid materials. The contact between surfaces did not mean closer contact with each other at every single point but clearance often filled by air. Compared to the surface with full contact, the air-gap caused by incomplete contact would not be helpful for heat conduction.

Figure 4 displays the thermal conductivity of different of buckypapers, where the parallel CNTs do not demonstrate aligned CNTs, but briefly display the structures without PVA. It should be noted that a spherical model was used to describe PVA, despite the fact that PVA is not a spherical particle. As PVA content rose, the thermal conductivities first increased and then decreased. In the absence of PVA, pristine MWCNT buckypapers showed large amounts of micron-sized pores and gaps, leading to low thermal conductivity of 1.394 W/m·k. As PVA content increased, PVA replaced air and occupied the gaps (Figure 4b). The latter led to narrowed gaps, intertube connection, and enhanced phonon transmission coefficient between surfaces. Accordingly, the thermal conductivity improved from 1.394 W/m·k to 2.009 W/m·k. At PVA content of 37.5 %, the thermal conductivity decreased to 1.509 W/m·k due to two aspects. First, the system could benefit from proper concentration of PVA; however, PVA could not be uniformly distributed in MWCNT buckypapers due to its intrinsic polymeric properties (Figure 3b). Excess PVA failed to increase phonon transmission efficiency but instead formed weak PVA layers intercalated between the CNT intertubes [45]. Second, CNT networks became unbalanced due to aggregated PVA, leading to uneven distribution of pores and stress (Figure 3b and Figure 4c).

The influence of hot press treatment on micro-structures of MWCNT buckypapers was mentioned in Section 3.2. With the influence of temperature and pressure, PVA would release the autologous thermal residual stresses, and stresses would be redistributed. The promoted PVA semi-crystalline structure produced integrated crystal boundaries, which led to a longer phonon mean free path inside the PVA. As a result, the densification of the CNT networks caused by the hot press at 70 °C played a crucial role in improving the thermal conductivities, promoting 2.473 W/m·k. A similar conclusion was reached in the study by Trakakis [46], where it was found that the thermal conductivity increased with decreasing porosity. In addition, the thermal conductivity of composites reinforced by CNTs was affected by many factors, such as defects, phonon scattering, filler type, structure, functionalization, alignment, and network formation [47].

#### 3.3.2. Finite Element Analysis

To figure out the potential applications of buckypaper in electric devices, a simple electronic device model was developed and coupled temperature-displacement finite element analyses were performed [48,49]. It was assumed that the electronic component module consisted of a basal plate and chips (Figure 5a). Subsequently, buckypapers were introduced into the joint parts of the electronic component module as lining plates (Figure 5b). The simulations were based on the following conditions: heat radiation and heat convection were ignored, while heat transfer was considered; the developed models were isotropic and homogeneous. A typical simulation in ABAQUS consists of six steps: (a) geometric modeling; (b) material property input; (c) analysis steps setup; (d) loading conditions setup (includes boundary conditions and predefined temperature field); (e) meshing; (f) simulation and output of the result. The ambient temperature was set to 25 °C, and the buckypapers possessed a thermal conductivity of 2.473 W/m k. The thermal conductivity of the basal plate (supposed to be made of epoxy resin) was 0.22 W/m k [50], and the densities of the buckypaper, basal plate, and chips were 0.21 g/cm^3^, 1.25 g/cm^3^, and 0.88 g/cm^3^, respectively. The specific heat capacities of the buckypaper, basal plate, and chips were 530 J/(kg·K), 1650 J/(kg·K), and 1800 J/(kg·K), respectively [51]. The software could output the temperature and stress field based on the heat transfer process and thermal expansion of each component in the model. After operating at a power of 150 W for 10 min, chips were heated up, and the overall temperature of the module was improved. The temperature and thermal stress distributions of the module were monitored, and the results are shown in Figure 6 and Figure 7, respectively.

Figure 6a–d demonstrates the temperature distributions on the front and reverse sides of the basal plate with or without buckypapers. The obtained maximum and minimum temperatures of the matrix were labeled on the images. Compared to the none-covered basal plate, the covered basal plate illustrated homogeneous temperature distributions. Meanwhile, the maximum temperature decreased rapidly from 97.3 °C to 90.5 °C and minimum temperature increased from 30.1 °C to 31.4 °C. This could be explained by the classical theory of heat conduction as follows. First, the buckypapers possessed a higher thermal conductivity; thus, they could receive more heat flow and spread along the in-plane direction quickly. Second, the buckypapers could fill in the air gaps between chips and basal plate, decreasing the thermal gradient between them. In addition, in the actual situation, buckypapers could possess adequate specific surface area which could be used for heat dissipation. Thus, buckypapers were able to quickly transmit excess heat from high-temperature regions to the surrounding regions preventing thermal concentration of the plate.

Figure 7a–d illustrates the thermal stress distributions on the front and reverse sides of the basal plate with and without buckypapers. The applied temperature field was the temperature distribution of Figure 6. The related mechanical parameters were collected from the work of Patole [35]. Due to the difference in the thermal expansion coefficient between the chips and basal plate, thermal stress would be generated at the contact location. In addition, it can be concluded that the higher the temperature, the greater the thermal stress. Clearly, the stress concentration at the contact position between the chip and the basal plate without buckypapers was much higher than that with buckypapers. Moreover, the thermal stress distribution of the basal plate with buckypapers was more homogeneous due to the uniform temperature distribution. Therefore, the existence of buckypapers would prevent the materials from the potential failure caused by localized overheating, thereby improving material security.

### 3.4. Electrical Conductivity and Electromagnetic Interference Shielding Properties

#### 3.4.1. Electrical Conductivity

The electrical performances of MWCNT/PVA buckypapers are shown in Figure 8a. As PVA content rose, the electrical conductivities of the buckypapers declined. This could be explained by the classic percolation theory. The classical statistical percolation model is applied to binary composite conductive systems, which consider the arrangement of conducting particles in a composite system as a regular arrangement of points or bonds in two or three dimensions. When the occupancy of conductive particles reaches a certain critical value, a conductive network is formed. In this study, the effect of PVA content on the electrical properties should be considered as the inverse process of the percolation theory mentioned above. As can be seen in Figure 3a, the MWCNT buckypaper with low PVA content possessed relatively perfect and continuous CNT networks with effective transmission paths of electrons identified as S1 (Figure 8b). In presence of PVA, the low conductive phase and the continuous structures became slightly disrupted due to the long-chain molecule of PVA immersed in inter-bundles. Therefore, the addition of more PVA content led to extension of the effective transmission paths for electrons (S2 > S1), leading to lower electrical conductivities at room temperature. Excess PVA could form an almost continuous network at high content (Figure 3b and Figure 8d). However, the electrical conductivities increased significantly after hot press treatment at 70 °C. With the action of pressure and temperature, the interface bonding between PVA and CNTs became stronger. In addition, the behavior of the thermal conductivity was similar to that of the electrical conductivity, which can be observed in Figure 8a. The electrical conductivities at room temperature improved due to the strengthening of CNT networks produced by the hot pressing method described in Section 3.2.

#### 3.4.2. Electromagnetic Interference Shielding Properties

The electromagnetic interference shielding could be defined as the logarithmic ratio of incoming power (Pi) to transmitted power (Pt) of an electromagnetic wave. In this study, electromagnetic waves were applied to the MWCNT/PVA buckypaper surface, and parameters like reflection (SER), absorption (SEA) and multiple reflection (SEMR) were collected. For reference, materials with 20 dB EMI shielding effectiveness (criteria for commercial applications) can shield 99% of electromagnetic waves [52,53].

The shielding effectiveness (SE) total would represent the total value contributed by reflection (SER), absorption (SEA), and multiple reflections (SEMR). It can be described according to Equation (1):
SE total = SER + SEA + SEMR(dB)(1)

At SE total ≥ 15 dB, SEMR can be neglected and SE total can be simplified by Equation (2):SE total ≅ SER + SEA(2)

In Figure 9, all parameters (SE total, SEA, and SER) showed a similar behavior with electrical conductivities, indicating that materials with good electrical conductivities would possess better EMI shielding abilities. Neat MWCNTs buckypapers showed the best average effectiveness of 27.56 dB when compared to other buckypapers at different PVA contents with the best EMI performance of 24.38 dB.

Electromagnetic shielding controls the propagation of electromagnetic waves from one region to another. When an electromagnetic wave propagates to the surface of a shielding material, there are usually three different attenuation mechanisms: (1) reflection attenuation caused by an impedance mismatch on the incident surface; (2) absorption attenuation inside the shielding material; and (3) multiple reflection attenuation in the shielding material. Neat MWCNTs buckypapers had the optimal specific area, which resulted in sufficient absorption and reflection attenuation. In addition, after functionalization, nitrogen- and oxygen-containing functional groups would form atoms on and off the carbon nanotube or graphene surface to enhance the EMI performance. The introduction of PVA into MWCNT networks lowered the electrical conductivities of buckypapers, leading to effective transmission paths for electrons, as explained in Section 3.4.1. In addition, materials with superior structural integrity should significantly improve EMI performance [54]. PVA would not only collapse the structural integrity of MWCNTs but reduce their conductivity to reach merely 10^−5^ S/m. Furthermore, the hot press method improved the EMI performance (Figure 9a). The neat buckypapers showed the best EMI performance of 31.79 dB. At 16.7 wt% PVA content, buckypaper exhibited an improved average effectiveness of 27.08 dB. Hot pressed buckypapers displayed firm electrical networks, as mentioned in Section 3.2, contributing to enhanced EMI performances due to increased electrical conductivities.

To further explore the electromagnetic shielding mechanism, the shielding effectiveness absorption and reflection were collected and the data are provided in Figure 9b,c. SEA and SER values of MWCNT/PVA buckypapers showed the same trend as SE total. However, the reflection showed no obvious change when compared to absorption.

Therefore, PVA and pressure would reduce the interface and porosity within MWCNT/PVA composites. The shielding effectiveness absorption was dominated by two phenomena: dielectric and magnetic losses. The dielectric loss played a more important role in the effectiveness of absorption due to the nonmagnetic characteristics of CNTs. Note, that dielectric loss should be influenced by polarization and conductivity, in which polarization would be derived from defects and interfaces. In Refs. [5,55,56], porous structures will affect the electromagnetic interference shielding properties due to an increase in internal numerous scattering and reflections, highly enhancing SEA.

## 4. Conclusions

MWCNT/PVA buckypapers were prepared by a vacuum-filtrating-hot press method to effectively improve the electrical conductivity, electromagnetic interference shielding properties, and thermal conductivity. The increase in PVA content led to enhanced thermal conductivity but reduced electrical conductivity and EMI performance. The electrical, shielding and thermal property mechanisms were discussed from the CNT network densification viewpoint. The finite element method was employed to verify the practical aspect of buckypapers in coupled temperature-displacement simulations. The developed electronic component model with buckypapers (2.473 W/m k) showed a more uniform surface temperature distribution with maximum temperature decreases of 97.3 °C to 90.5 °C. Meanwhile, the minimum temperature increased from 30.1 °C to 31.4 °C. In addition, the thermal stress distribution of the model with buckypapers possessed no obvious stress concentration. In a short, the buckypapers prevented the basal plate from aging and fatigue at high temperatures, and extended the service life of materials. Compared to MWCNTs, SWCNTs and DWCNTs exhibited better performances concerning the thermal and electrical conductivities, which can provide a potential for multifunctional buckypapers. However, SWCNTs and DWCNTs would be difficult to disperse in an aqueous solution due to their thinner and fewer tubes. The proposed method of fabricating MWCNT/PVA buckypapers looks promising for the production of flexible and multifunctional materials following a fast and cost-effective approach.

## Figures and Tables

**Figure 1 nanomaterials-10-02503-f001:**
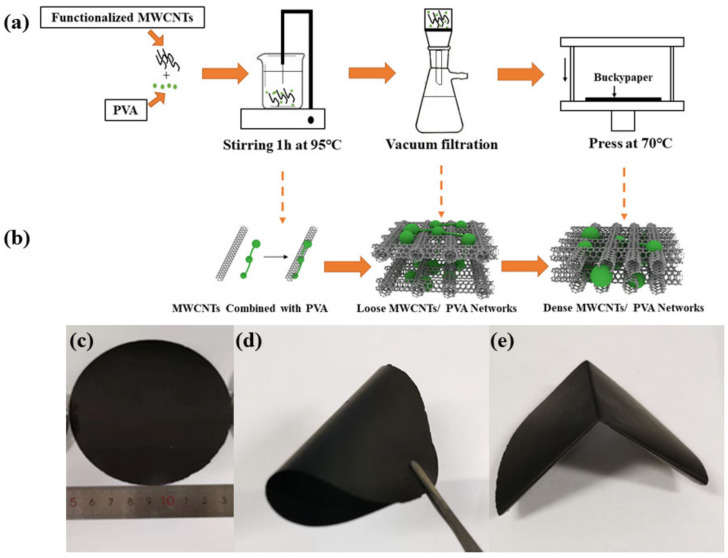
Experimental set-up and results. (**a**) Schematic of the preparation of multi-walled carbon nanotubes (MWCNT)/polyvinyl alcohol (PVA) buckypapers; (**b**) microstructure evolution of MWCNTs and buckypapers during the preparation process; (**c**–**e**) photographs of the MWCNT/PVA buckypapers.

**Figure 2 nanomaterials-10-02503-f002:**
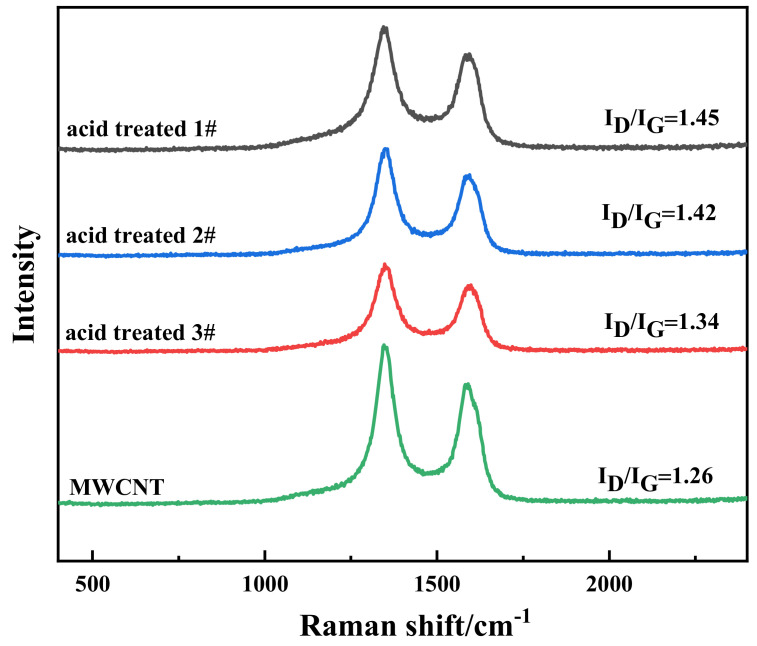
Raman spectra of pristine MWCNTs and acidification MWCNTs with excitation at 532 nm.

**Figure 3 nanomaterials-10-02503-f003:**
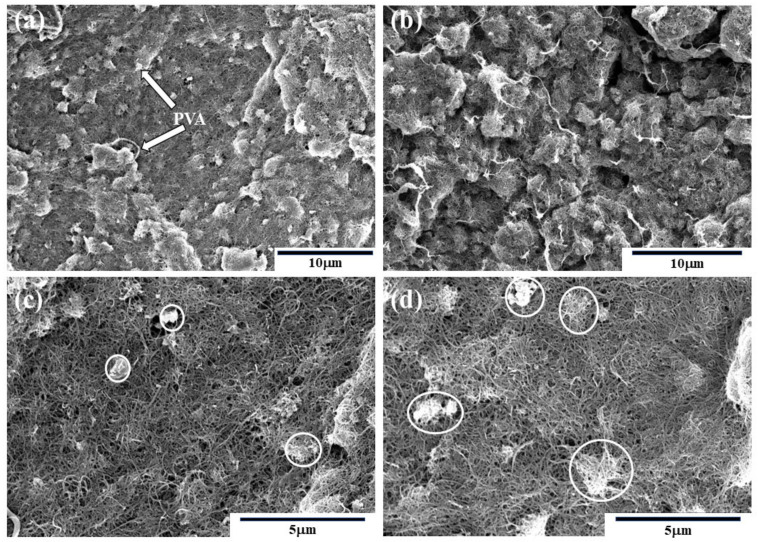
SEM images of MWCNT/PVA buckypapers. (**a**) 5.7 wt% PVA; (**b**) 37.5 wt% PVA, (**c**) 5.7 wt% PVA after hot pressing; (**d**) 37.5 wt% PVA after hot pressing.

**Figure 4 nanomaterials-10-02503-f004:**
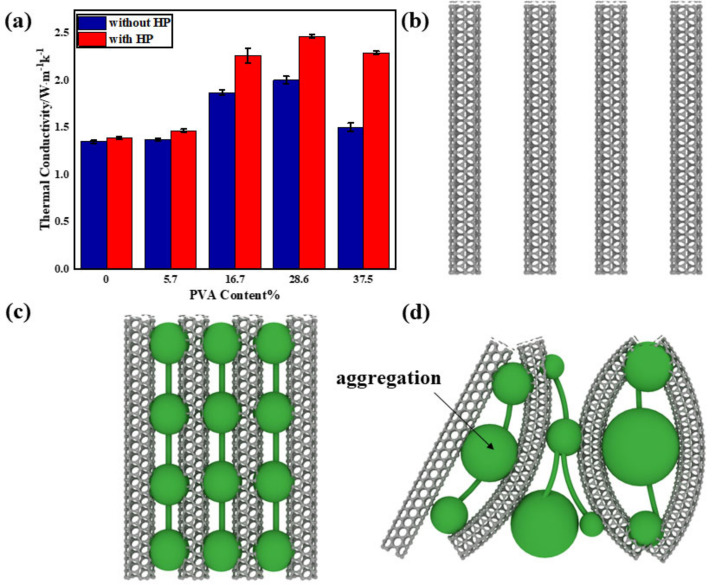
(**a**) Thermal conductivity of buckypapers at different PVA contents. (**b**–**d**) Microstructure evolution of MWCNT networks as a function of PVA content.

**Figure 5 nanomaterials-10-02503-f005:**
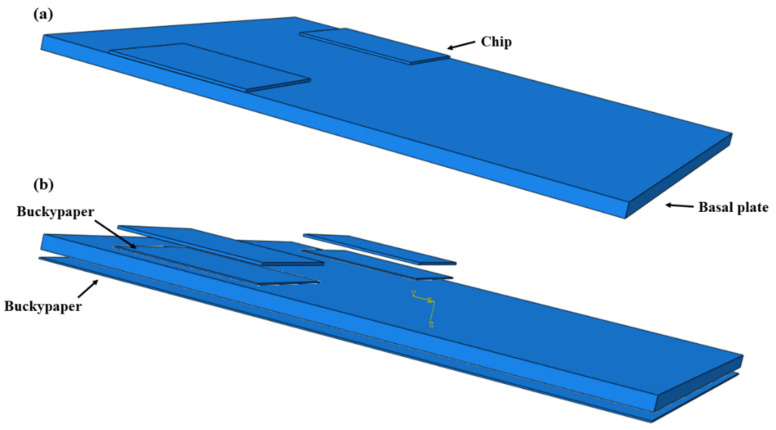
Model diagram of the electronic component module, (**a**) model without buckypapers, (**b**) model with buckypapers.

**Figure 6 nanomaterials-10-02503-f006:**
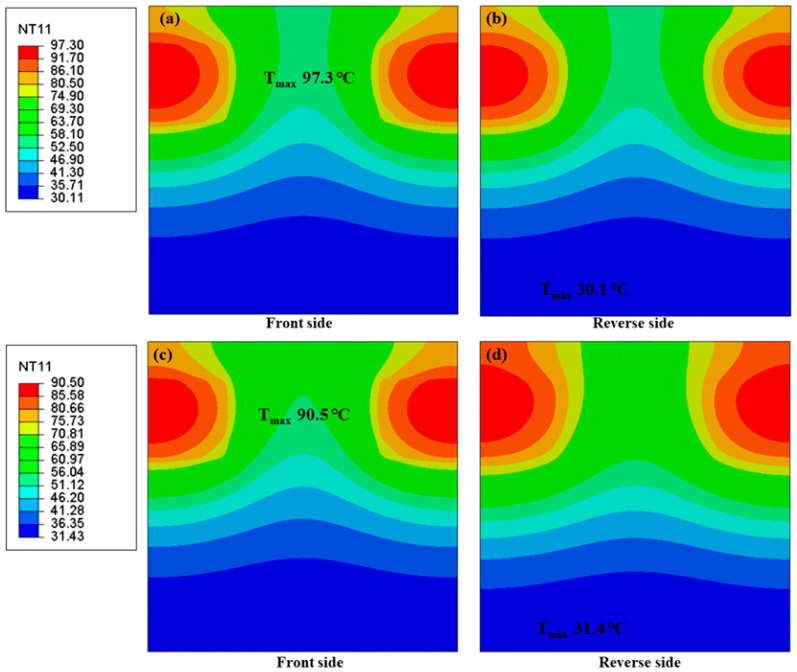
Temperature distribution maps of the heat transfer model without and with buckypaper. (**a**,**b**) Without buckypaper; (**c**,**d**) with buckypaper of 2.4732 W/m·k.

**Figure 7 nanomaterials-10-02503-f007:**
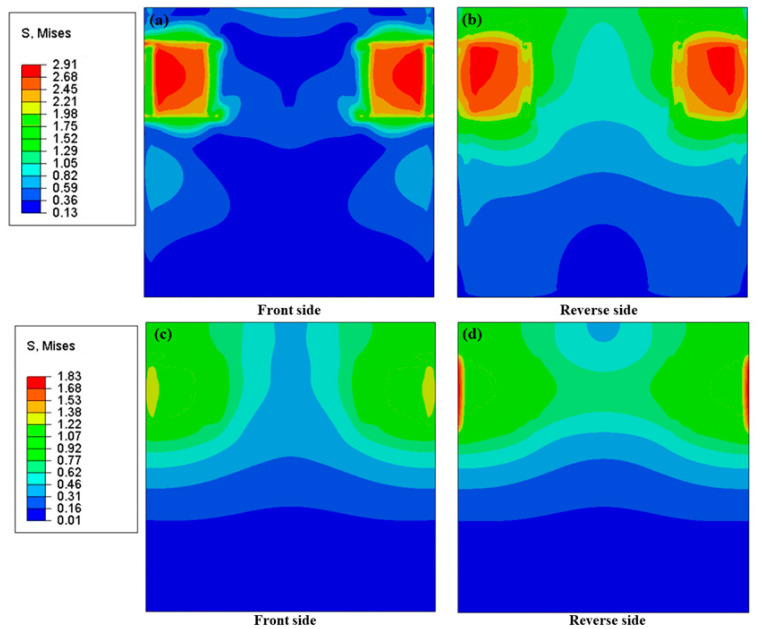
Thermal stress distribution maps of the heat transfer model without and with buckypaper. (**a**,**b**) Without buckypaper; (**c**,**d**) with buckypaper of 2.4732 W/m·k.

**Figure 8 nanomaterials-10-02503-f008:**
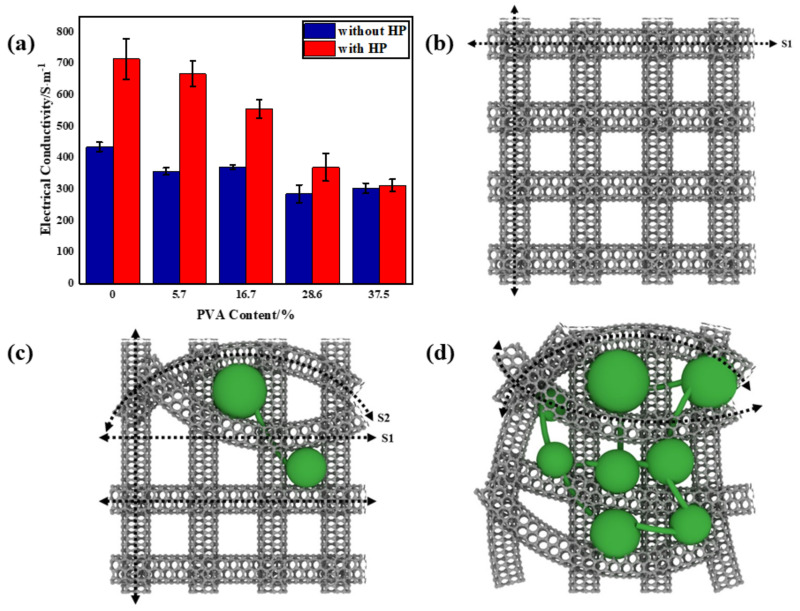
(**a**) Electrical conductivity of buckypapers at different PVA contents. (**b**–**d**) Microstructure evolution of MWCNT networks as a function of PVA content.

**Figure 9 nanomaterials-10-02503-f009:**
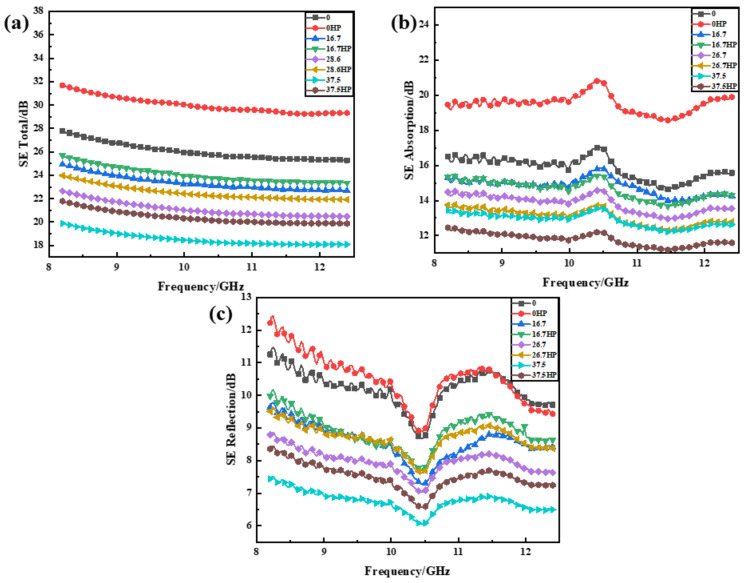
Electromagnetic interference (EMI) shielding effectiveness (SE) versus frequency of MWCNT/PVA buckypapers. (**a**) SE total, (**b**) SE adsorption, and (**c**) SE reflection.

**Table 1 nanomaterials-10-02503-t001:** Dimension parameters of buckypapers.

Sample	Pore Diameter/nm		Pore Area/nm^2^		Thickness/mm		Density/g·cm^−3^	
	Before	After	Before	After	Before	After	Before	After
BP 0#	25.80	23.56	546.95	512.01	0.28	0.12	0.20	0.46
BP 5.7#	22.38	20.49	504.03	488.36	0.25	0.11	0.22	0.50
BP 16.7#	22.12	19.22	454.24	402.80	0.25	0.08	0.22	0.69
BP 28.6#	20.15	19.36	413.19	394.45	0.24	0.12	0.23	0.46
BP 37.5#	18.96	17.02	395.02	383.66	0.26	0.10	0.21	0.55

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
