# Peer review of "Multi-Functional Properties of MWCNT/PVA Buckypapers Fabricated by Vacuum Filtration Combined with Hot Press: Thermal, Electrical and Electromagnetic Shielding"

_nanomaterials, 2020, doi:10.3390/nano10122503_

Round 1

Reviewer 1 Report

The modifications are proper and I feel that the article is suitable for publication.

Author Response

Thanks to the editor and reviewer for the comments.

Reviewer 2 Report

The authors did a revision of their manuscript. However they did not answer to some question raised in the previous report. The discussion could be improved adding more information and comparing results with current literature.

    1.  

Revise the title, the buckypaper does not offer any thermal shielding as authors show in their work.

    1.  

Row 89: Preparation of bukypaper. The description of the buckypaper synthesis is unclear. Please better describe the synthesis method: preprocessing: preparation of functionalized MWCNT. Buckypaper synthesis: two hot-press stages are indicated why? Two routes are indicated in the scheme fig 1a please specify why two different routes are required to obtain the buckypaper. Also hot press and vulcanizing terms are used to identify the same process. Please uniform the nomenclature throughout the manuscript text.

Row 162 PVA glass transition. The glass transition temperature of PVA is 75 C. Why the authors performed the hot pressing at 70 oC? Properties of the buckypaper likely depend on the temperature of the hot-press process. Authors must characterize the buckypaper as a function of the hot-press temperature.

Row 195: authors placed the buckypaper just under the chips and on the whole backside of the epoxy base. (i) they should justify why the buckypaper was placed just under chips and not extended on the whole epoxy base since problems due to electrical connections affect equally both the front and back side of the base.

Figure 6 a, b: in absence of buckypaper the thermal distribution of the front and back side of the base are very similar. Considering that the expoxy has a thermal conductivity which is about 10 times lower than that of the buckypaper one expects that the backside of the base should show a lower temperature with respect to the front side. Why they are equal? Figure 6 c, d: in presence of buckypaper, why the back side of the base shows a distribution of high temperature more extended than that of the front side?

Row 302: authors state that “after functionalization, nitrogen-containing, oxygen-containing functional groups would form atoms on and off the carbon nanotube or graphene surface to enhance electrical and EMI performances”. It is well known that presence of oxygen and nitrogen functional groups are detrimental for the electrical conductivity.

Row 286: acronym SE not defined. SEM, SER, SEMR are defined in rows 282, 283. Attention to definition of SEMR

English needs revision.

Round 2

Reviewer 2 Report

//

This manuscript is a resubmission of an earlier submission. The following is a list of the peer review reports and author responses from that submission.

Round 1

Reviewer 1 Report

The paper presents fabrication of acid treated MWCNTs/PVA buckypapers by vacuum filtration and hot pressing. The thermal, electrical and EM shielding properties of the compounds were studied. The paper is interesting but major revisions are necessary to meet Journal standards.

Similar literature works to fabricate backypapers (indicatevely http://dx.doi.org/10.1016/j.cej.2015.06.085, http://dx.doi.org/10.1016/j.compscitech.2013.01.003 ) should be mentioned along with a discussion about the advantages of the proposed method.           

2. The acid treatment is a well-known process and its effect on MWCNTs integrity must be discussed in relation to the literature.

3. In the Raman spectra of pristine and acid treated MWCNTs a mode softening is observed. This finding should be discussed. Also, the increase in the I(D/G) ratio is minor and, in my opinion, is not a convinced argument. At least the authors should provide additional Raman measurements from different sample points showing that the same trend holds for the mean values. Another alternative technique such as XPS technique should be considered.

4. sp2 -sp3 typo

5. The thermal conductivity performance should be compared and discussed with analogous literature works (indicatevely Materials 2020, 13, 4308; doi:10.3390/ma13194308).

6. cm3 typo

7. Technical information concerning Finite element calculations should be given (in methods) in case  someone wants to reproduce the findings.

8. The porosity parameters, based on the image analysis of SEM images, give an estimate of the surface porosity which is not intrinsic property. Therefore, the values of pore diameterr and area in Table 1 are not quite reliable. Some other bulk porosity technique (e.g Hg intrusion porosimetry) to extract those values should be used.

Reviewer 2 Report

The authors developed a method to synthesize buckypapers with improved physical properties. Considering the applicative perspectives of the material the characterization of the mechanical properties is mandatory as well as the change of the material properties as a function of the temperature applied during the hot-press process. In addition, the discussion should be improved adding more information and comparing results with current literature.

1) Row 89: Preparation of bukypaper. The description of the buckypaper synthesis does not correspond to the scheme. (i) as it appears from the scheme, MWCNT/PVA is prepared. Then, there are two routes to perform the hot pressing. Only one of these is described in the text. Please describe also the second route or eliminate it. (ii) The hot pressing temperature is 70 oC but 60 oC are indicated in the figure. Please verify.

2) Row 121: MWCNT acid treatment. The acid treatment has a variety of effects. It attacks the MWCNT surface at defects oxidizing the defect carbon atoms. The oxidation proceeds increasing the defect dimensions thus lowering the mechanical properties of the MWCNT and can end up with cutting the MWCNT. Also the electrical properties are affected by oxidation because threee different effects: (i) presence of bonded oxygen strongly reduces the conductivity MWCNT; (ii) presence of defects lowers the charge mobility; (iii) reduction of MWCNT length. Please better describe the effects of acid induced oxidation in the text

3) Row 128: Raman spectroscopy. The virgin and treated Raman spectra show bands slightly shifted each other. Raman bands are sensitive to carbon atom oxidation. Please better discuss the results comparing with literature.

4) Row 149 PVA glass transition. Authors verified that the glass transition temperature of PVA is 75 C. Why the authors performed the hot pressing at 70 oC? Higher temperature should facilitate the penetration of the PVA and reduce PVA aggregation. Dependence of buckypaper properties as a function of the hot-press temperature is mandatory.

5) Row 195: theoretical model. The description of the modeled system is not clear. (i) it is not clear the presence of a basal plane supposed to be of epoxy resin. A motivation for this is lacking. Then two sheets of buckypaper are placed below the chips (figure 5(b) while authors stated they used buckypaper to join the chips. What does it mean “join chips”? Why buckypaper is placed also below the basal plane? Provide more information and motivations.

6) Row 245. Electrical conductivity. If authors are interested in the electrical conductivity, why they do not add a ionic component to PVA to increase the charge propagation inside PVA? There are works in literature describing this strategy to improve PVA conductivity.

7) English needs revision.

Reviewer 3 Report

This research demonstrated the MWCNT/PVA buckypapers using PVA polymer and MWCNT suspension by the vacuum infiltration combined with hot press method. The results presented the enhanced thermal, electrical, and electromagnetic shielding properties because of dense MWCNT structures. It is important performance for the publication in Nanomaterials. It can published in Nanomaterials after simple revisions, as below. 1. Is this approach also useful for 2D carbon materials, such as graphene or reduced graphene oxide (RGO)? Why did the author select CNT instead of graphene in this study? Mention these contents in the manuscript, with additional high-impact references to attract broader readership as below. - Particle & Particle Systems Characterization, 34, 1600429 (2017) - ACS Nano, 13, 9332 (2019) - etc.